# COMPRESSED SENSING WITH DEEP IMAGE PRIOR AND LEARNED REGULARIZATION

## ABSTRACT

We propose a novel method for compressed sensing recovery using untrained deep generative models. Our method is based on the recently proposed Deep Image Prior (DIP), wherein the convolutional weights of the network are optimized to match the observed measurements. We show that this approach can be applied to solve any differentiable linear inverse problem, outperforming previous unlearned methods. Unlike various learned approaches based on generative models, our method does not require pre-training over large datasets. We further introduce a novel learned regularization technique, which incorporates prior information on the network weights. This reduces reconstruction error, especially for noisy measurements. Finally we prove that, using the DIP optimization approach, moderately overparameterized single-layer networks trained can perfectly fit any signal despite the nonconvex nature of the fitting problem. This theoretical result provides justification for early stopping.

## 1 INTRODUCTION

We consider the well-studied compressed sensing problem of recovering an unknown signal $x^* \in \mathbb{R}^n$ by observing a set of noisy measurements $y \in \mathbb{R}^m$ of the form

$$y = Ax^* + \eta. \tag{1}$$

Here $A \in \mathbb{R}^{m \times n}$ is a known measurement matrix, typically generated with random independent Gaussian entries. Since the number of measurements $m$ is smaller than the dimension $n$ of the unknown vector $x^*$, this is an under-determined system of noisy linear equations and hence ill-posed. There are many solutions, and some structure must be assumed on $x^*$ to have any hope of recovery. Pioneering research (Donoho, 2006; Candès et al., 2006; Candes & Tao, 2005) established that if $x^*$ is assumed to be sparse in a known basis, a small number of measurements will be provably sufficient to recover the unknown vector in polynomial time using methods such as Lasso (Tibshirani, 1996).

Sparsity approaches have proven successful, but more complex models with additional structure have been recently proposed such as model-based compressive sensing (Baraniuk et al., 2010) and manifold models (Hegde et al., 2008; Hegde & Baraniuk, 2012; Eftekhari & Wakin, 2015). Bora et al. (2017) showed that deep generative models can be used as excellent priors for images. They also showed that backpropagation can be used to solve the signal recovery problem by performing gradient descent in the generative latent space. This method enabled image generation with significantly fewer measurements compared to Lasso for a given reconstruction error. Compressed sensing using deep generative models was further improved in very recent work (Tripathi et al., 2018; Grover & Ermon, 2018a; Kabkab et al., 2018; Shah & Hegde, 2018; Fletcher & Rangan, 2017; Asim et al., 2018). Additionally a theoretical analysis of the nonconvex gradient descent algorithm (Bora et al., 2017) was proposed by Hand & Voroninski (2017) under some assumptions on the generative model.

Inspired by these impressive benefits of deep generative models, we chose to investigate the application of such methods for medical imaging, a canonical application of compressive sensing. A significant problem, however, is that all these previous methods require the existence of *pre-trained* models. While this has been achieved for various types of images, e.g. human faces of CelebA (Liu et al., 2015) via DCGAN (Radford et al., 2015), it remains significantly more challenging for med-

ical images (Wolterink et al., 2017; Schlegl et al., 2017; Nie et al., 2017; Schlemper et al., 2017). Instead of addressing this problem in generative models, we found an easier way to circumvent it.

Surprising recent work by Ulyanov et al. (2017) proposed Deep Image Prior (DIP), which uses *untrained* convolutional neural networks. In DIP-based schemes, a convolutional neural network generator (e.g. DCGAN) is initialized with random weights; these weights are subsequently optimized to make the network produce an output as close to the target image as possible. This procedure is unlearned, using no prior information from other images. The prior is enforced only by the fixed convolutional structure of the generator network.

Generators used for DIP are typically over-parameterized, i.e. the number of network weights is much larger compared to the output dimension. For this reason DIP has empirically been found to overfit to noise if run for too many iterations (Ulyanov et al., 2017). In this paper we theoretically prove that this phenomenon occurs with gradient descent and justify the use of early stopping and other regularization methods.

**Our Contributions:**

- In Section 3 we propose DIP for compressed sensing (CS-DIP). Our basic method is as follows. Initialize a DCGAN generator with random weights; use gradient descent to optimize these weights such that the network produces an output which agrees with the observed measurements as much as possible. This unlearned method can be improved with a novel *learned regularization* technique, which regularizes the DCGAN weights throughout the optimization process.

- In Section 4 we theoretically prove that DIP will fit any signal to zero error with gradient descent. Our result is established for a network with a single hidden layer and sufficient constant fraction over-parametrization. While it is expected that over-parametrized neural networks can fit any signal, the fact that gradient descent can provably solve this non-convex problem is interesting and provides theoretical justification for early stopping.

- In Section 5 we empirically show that CS-DIP outperforms previous unlearned methods in many cases. While pre-trained or "learned" methods frequently perform better (Bora et al., 2017), we have the advantage of not requiring a generative model trained over large datasets. As such, we can apply our method to various medical imaging datasets for which data acquisition is expensive and generative models are difficult to train.

## 2 BACKGROUND

### 2.1 COMPRESSED SENSING: CLASSICAL AND UNLEARNED APPROACHES

A classical assumption made in compressed sensing is that the vector $x^*$ is $k$-sparse in some basis such as wavelet or discrete cosine transform (DCT). Finding the sparsest solution to an underdetermined linear system of equations is NP-hard in general; however, if the matrix $A$ satisfies conditions such as the Restricted Eigenvalue Condition (REC) or Restricted Isometry Property (RIP) (Candes et al., 2006; Bickel et al., 2009; Donoho, 2006; Tibshirani, 1996), then $x^*$ can be recovered in polynomial time via convex relaxations (Tropp, 2006) or iterative methods. There is extensive compressed sensing literature regarding assumptions on $A$, numerous recovery algorithms, and variations of RIP and REC (Bickel et al., 2009; Negahban et al., 2009; Agarwal et al., 2010; Bach et al., 2012; Loh & Wainwright, 2011).

Compressed sensing methods have found many applications in imaging, for example the single-pixel camera (SPC) (Duarte et al., 2008). Medical tomographic applications include x-ray radiography, microwave imaging, magnetic resonance imaging (MRI) (Winters et al., 2010; Chen et al., 2008; Lustig et al., 2007). Obtaining measurements for medical imaging can be costly, time-consuming, and in some cases dangerous to the patient (Qaisar et al., 2013). As such, an important goal is to reduce the number of measurements while maintaining good reconstruction quality.

Aside from the classical use of sparsity, recent work has used other priors to solve linear inverse problems. Plug-and-play priors (Venkatakrishnan et al., 2013; Chan et al., 2017) and Regularization by Denoising (Romano et al., 2017) have shown how image denoisers can be used to solve general linear inverse problems. A key example of this is BM3D-AMP, which applies a Block-Matching

and 3D filtering (BM3D) denoiser to an Approximate Message Passing (D-AMP) algorithm (Metzler et al., 2016; 2015). AMP has also been applied to linear models in other contexts (Schniter et al., 2016). Another related algorithm is TVAL3 (Zhang et al., 2013; Li et al., 2009) which leverages augmented Lagrangian multipliers to achieve impressive performance on compressed sensing problems. In many different settings, we compare our algorithm to these prior methods: BM3D-AMP, TVAL3, and Lasso.

## 2.2 COMPRESSED SENSING: LEARNED APPROACHES

While sparsity in some chosen basis is well-established, recent work has shown better empirical performance when neural networks are used (Bora et al., 2017). This success is attributed to the fact that neural networks are capable of learning image priors from very large datasets (Goodfellow et al., 2014; Kingma & Welling, 2013). There is significant recent work on solving linear inverse problems using various learned techniques, e.g. recurrent generative models (Mardani et al., 2017b) and auto-regressive models (Dave et al., 2018). Additionally approximate message passing (AMP) has been extended to a learned setting by Metzler et al. (2017).

Bora et al. (2017) is the closest to our set-up. In this work the authors assume that the unknown signal is in the range of a pre-trained generative model such as a generative adversarial network (GAN) (Goodfellow et al., 2014) or variational autoencoder (VAE) (Kingma & Welling, 2013). The recovery of the unknown signal is obtained via gradient descent in the latent space by searching for a signal that satisfies the measurements. This can be directly applied for linear inverse problems and more generally to any differentiable measurement process. Recent work has built upon these methods using new optimization techniques (Chang et al., 2017), uncertainty autoencoders (Grover & Ermon, 2018b), and other approaches (Dhar et al., 2018; Kabkab et al., 2018; Mixon & Villar, 2018; Pandit et al., 2019; Rusu et al., 2018; Hand et al., 2018). The key point is that all this prior work requires pre-trained generative models, in contrast to CS-DIP. Finally, there is significant ongoing work to understand DIP and develop related approaches (Heckel et al., 2018; Dittmer et al., 2018).

## 3 PROPOSED ALGORITHM

Let $x^* \in \mathbb{R}^n$ be the signal that we are trying to reconstruct, $A \in \mathbb{R}^{m \times n}$ be the measurement matrix, and $\eta \in \mathbb{R}^m$ be independent noise. Given the measurement matrix $A$ and the observations $y = Ax^* + \eta$, we wish to reconstruct an $\hat{x}$ that is close to $x^*$.

A generative model is a deterministic function $G(z; w)$: $\mathbb{R}^k \to \mathbb{R}^n$ which takes as input a seed $z \in \mathbb{R}^k$ and is parameterized by "weights" $w \in \mathbb{R}^d$, producing an output $G(z; w) \in \mathbb{R}^n$. These models have shown excellent performance generating real-life signals such as images (Goodfellow et al., 2014; Kingma & Welling, 2013) and audio (Van Den Oord et al., 2016). We investigate *deep convolutional* generative models, a special case in which the model architecture has multiple cascaded layers of convolutional filters (Krizhevsky et al., 2012). In this paper we apply a DC-GAN (Radford et al., 2015) model and restrict the signals to be images.

## 3.1 COMPRESSED SENSING WITH DEEP IMAGE PRIOR (CS-DIP)

Our approach is to find a set of weights for the convolutional network such that the measurement matrix applied to the network output, i.e. $AG(z; w)$, matches the measurements $y$ we are given. Hence we initialize an *untrained* network $G(z; w)$ with some fixed $z$ and solve the following:

$$w^* = \arg\min_w \|y - AG(z; w)\|^2. \tag{2}$$

This is, of course, a non-convex problem because $G(z; w)$ is a complex feed-forward neural network. Still we can use gradient-based optimizers for any generative model and measurement process that is differentiable. Generator networks such as DCGAN are biased toward smooth, natural images due to their convolutional structure; thus the network structure alone provides a good prior for reconstructing images in problems such as inpainting and denoising (Ulyanov et al., 2017). Our finding is that this applies to general linear measurement processes. Furthermore, our method also directly applies to any differentiable forward operator $A$. We restrict our solution to lie in the span

of a convolutional neural network. If a sufficient number of measurements $m$ is given, we obtain an output such that $x^* \approx G(z; w^*)$.

Note that this method uses an untrained generative model and optimizes over the network weights $w$. In contrast previous methods, such as that of Bora et al. (2017), use a trained model and optimize over the latent $z$-space, solving $z^* = \arg\min_z \|y - AG(z; w)\|^2$. We instead initialize a random $z$ with Gaussian i.i.d. entries and keep this fixed throughout the optimization process.

In our algorithm we leverage the well-established total variation regularization (Rudin et al., 1992; Wang et al., 2008; Liu et al., 2018), denoted as $TV(G(z; w))$. We also propose an additional learned regularization technique, $LR(w)$; note that without this technique, i.e. when $\lambda_L = 0$, our method is completely unlearned. Lastly we use early stopping, a phenomena that will be analyzed theoretically in Section 4.

Thus the final optimization problem becomes

$$w^* = \arg\min_w \|y - A\,G(z; w)\|^2 + R(w; \lambda_T, \lambda_L). \tag{3}$$

The regularization term contains hyperparameters $\lambda_T$ and $\lambda_L$ for total variation and learned regularization: $R(w; \lambda_T, \lambda_L) = \lambda_T TV(G(z; w)) + \lambda_L LR(w)$. Next we discuss this $LR(w)$ term.

## 3.2 LEARNED REGULARIZATION

Without learned regularization CS-DIP relies only on linear measurements taken from one unknown image. We now introduce a novel method which leverages a small amount of training data to optimize regularization. In this case training data refers to measurements from additional ground truth of a similar type, e.g. other x-ray images.

To leverage this additional information, we pose Eqn. 3 as a Maximum a Posteriori (MAP) estimation problem and propose a novel prior on the weights of the generative model. This prior then acts as a regularization term, penalizing the model toward an optimal set of weights $w^*$.

For a set of weights $w \in \mathbb{R}^d$, we model the *likelihood* of the measurements $y = Ax, y \in \mathbb{R}^m$, and the prior on the weights $w$ as Gaussian distributions given by

$$p(y|w) = \frac{\exp\left(-\dfrac{\|y - AG(z; w)\|^2}{2\lambda_L}\right)}{\sqrt{(2\pi\lambda_L)^m}}; \qquad p(w) = \frac{\exp\left(-\frac{1}{2}(w - \mu)^T \Sigma^{-1}(w - \mu)\right)}{\sqrt{(2\pi)^d|\Sigma|}},$$

where $\mu \in \mathbb{R}^d$ and $\Sigma \in \mathbb{R}^{d \times d}$.

In this setting we want to find a set of weights $w^*$ that maximizes the posterior on $w$ given $y$, i.e.,

$$w^* = \arg\max_w p(w|y) \equiv \arg\min_w \|y - AG(z; w)\|^2 + \lambda_L (w - \mu)^T \Sigma^{-1} (w - \mu). \tag{4}$$

This gives us the learned regularization term

$$LR(w) = (w - \mu)^T \Sigma^{-1} (w - \mu), \tag{5}$$

where the coefficient $\lambda_L$ in Eqn. 4 controls the strength of the prior.

Our motivation for assuming a Gaussian distribution on the weights is to build upon the proven success of $\ell_2$ regularization, which also makes this assumption. Notice that when $\mu = 0$ and $\Sigma = I_{d \times d}$, this regularization term is equivalent to $\ell_2$-regularization. Thus this method can be thought of as an adaptive version of standard weight decay. Further, because the network weights are initialized Gaussian i.i.d., we assumed the optimized weights would also be Gaussian. Previous work has shown evidence that the convolutional weights in a trained network do indeed follow a Gaussian distribution (Ma et al., 2018).

### 3.2.1 LEARNING THE PRIOR PARAMETERS

In the previous section, we introduced the learned regularization term $LR(w)$ defined in Eqn. 5. However we have not yet learned values for parameters $(\mu, \Sigma)$ that incorporate prior knowledge of the network weights. We now propose a way to estimate these parameters.

Assume we have a set of measurements $S_Y = \{y_1, y_2, \cdots, y_Q\}$ from $Q$ different images $S_X = \{x_1, x_2, \cdots, x_Q\}$, each obtained with a different measurement matrix $A$. For each measurement $y_q, q \in \{1, 2, ..., Q\}$, we run CS-DIP to solve the optimization problem in Eqn. 3 and obtain an optimal set of weights $W^* = \{w_1^*, w_2^*, \cdots, w_Q^*\}$. Note that when optimizing for the weights $W^*$, we only have access to the measurements $S_Y$, not the ground truth $S_X$.

The number of weights $d$ in deep networks tends to be very large. As such, learning a distribution over each weight, i.e. estimating $\mu \in \mathbb{R}^d$ and $\Sigma \in \mathbb{R}^{d \times d}$, becomes intractable. We instead use a layer-wise approach: with $L$ network layers, we have $\mu \in \mathbb{R}^L$ and $\Sigma \in \mathbb{R}^{L \times L}$. Thus each weight within layer $l \in \{1, 2, ..., L\}$ is modeled according to the same $\mathcal{N}(\mu_l, \Sigma_{ll})$ distribution. For simplicity we assume $\Sigma_{ij} = 0 \; \forall \, i \neq j$, i.e. that network weights are independent across layers. The process of estimating statistics $(\mu, \Sigma)$ from $W^*$ is described in Algorithm 1 of the appendix.

We use this learned $(\mu, \Sigma)$ in the regularization term $LR(w)$ from Eqn. 5 for reconstructing measurements of images. We refer to this technique as *learned regularization*. While this may seem analogous to batch normalization (Ioffe & Szegedy, 2015), note that we only use $(\mu, \Sigma)$ to penalize the $\ell_2$-norm of the weights and do not normalize the layer outputs themselves.

### 3.2.2 DISCUSSION OF LEARNED REGULARIZATION

The proposed CS-DIP does not require training if no learned regularization is used, i.e. if $\lambda_L = 0$ in Eqn. 3. This means that CS-DIP can be applied only with measurements from a single image and no prior information of similar images in a dataset.

Our next idea, learned regularization, utilizes a small amount of prior information, requiring access to measurements from a small number of similar images (roughly $5 - 10$). In contrast, other pre-trained models such as that of Bora et al. (2017) require access to ground truth from a massive number of similar images (tens of thousands for CelebA). If such a large dataset is available, and if a good generative model can be trained on that dataset, we expect that pre-trained models would outperform our method. Our approach is instead more suitable for reconstructing problems where large amounts of data or good generative models are not readily available.

## 4 THEORETICAL RESULTS

In this section we provide theoretical evidence to highlight the importance of early stopping for DIP-based approaches. Here we focus on denoising a noisy signal $y \in \mathbb{R}^n$ by optimizing over network weights. This problem takes the form:

$$\min_{w} \; \mathcal{L}(w) := \|y - G(z; w)\|^2. \tag{6}$$

This is a special instance of Eqn. 2 with the measurement matrix $A = I$ corresponding to denoising. We focus on generators consisting of a single hidden-layer ReLU network with $k$ inputs, $d$ hidden units, and $n$ outputs. Using $w = (W, V)$ the generator model in this case is given by

$$G(z; W, V) = V \cdot \text{ReLU}(Wz), \tag{7}$$

where $z \in \mathbb{R}^k$ is the input, $W \in \mathbb{R}^{d \times k}$ the input-to-hidden weights, and $V \in \mathbb{R}^{n \times d}$ the hidden-to-output weights. We assume $V$ is fixed at random and train over $W$ using gradient descent. With these formulations in place, we are now ready to state our theoretical result.

**Theorem 4.1.** *Consider fitting a generator of the form $W \mapsto G(z; W, V) = V \cdot ReLU(Wz)$ to a signal $y \in \mathbb{R}^n$ with $z \in \mathbb{R}^k$, $W \in \mathbb{R}^{d \times k}$, $V \in \mathbb{R}^{n \times d}$, and $ReLU(z) = \max(0, z)$. Furthermore, assume $V$ is a random matrix with i.i.d. $\mathcal{N}(0, \nu^2)$ entries with $\nu = \frac{1}{\sqrt{dn}} \frac{\|y\|}{\|z\|}$. Starting from an initial weight matrix $W_0$ selected at random with i.i.d. $\mathcal{N}(0, 1)$ entries, we run gradient descent updates of the form $W_{\tau+1} = W_\tau - \eta \nabla \mathcal{L}(W_\tau)$ on the loss*

$$\mathcal{L}(W) = \frac{1}{2} \|V \cdot ReLU(Wz) - y\|^2,$$

*with step size $\eta = \frac{\bar{\eta}}{\|y\|^2} \frac{8n}{4n+d}$ where $\bar{\eta} \leq 1$. Assuming that*

$$d \geq Cn,$$

*with $C$ a fixed numerical constant, then*

$$\|V \cdot ReLU(W_\tau z) - y\| \leq 3 \left(1 - \frac{\bar{\eta}}{8(4n+d)}\right)^\tau \|y\|$$

*holds for all $\tau$ with probability at least $1 - 5e^{-n/2} - e^{-d/2} - e^{-4d^{\frac{2}{3}}n^{\frac{1}{3}}}$.*

Our theoretical result shows that after many iterative updates, gradient descent will solve this non-convex optimization problem and fit any signal $y$, if the generator network is sufficiently wide. This occurs as soon as the number of hidden units $d$ exceeds the signal size $n$ by a constant factor. In our theorem we have focused on the case where the measurement matrix is the identity ($A = I$). We note however that our theorem directly applies to many compressed sensing measurement matrices, in particular any matrix obtained by subsampling the rows of an orthonormal matrix (e.g. sub-sampling a Fourier matrix). This is possible as for any such matrix, $AV$ has the same distribution as a Gaussian matrix with i.i.d. entries. Therefore, our theorem directly applies in this setting by replacing $V$ with $AV$. This result demonstrates that early stopping is necessary for DIP-based methods to be successful; otherwise the network can fit any signal, including one that is noisy.

Our proof builds on theoretical ideas from Oymak & Soltanolkotabi (2019) which provide a general framework for establishing global convergence guarantees for overparameterized nonlinear learning problems based on various properties of the Jacobian mapping along the gradient descent trajectory. Related literature can be found in Du et al. (2018); Oymak & Soltanolkotabi (2018). We combine tools from empirical process theory, random matrix theory, and matrix algebra to show that, starting from a random initialization, the Jacobian mapping across all iterates has favorable properties with high probability, hence facilitating convergence to a global optima.

## 5 EXPERIMENTS

### 5.1 EXPERIMENTAL SETUP

**Measurements:** We evaluate our algorithm using two different measurements processes, i.e. matrices $A \in \mathbb{R}^{m \times n}$. First we set the entries of $A$ to be Gaussian i.i.d. such that $A_{i,j} \sim \mathcal{N}(0, \frac{1}{m})$. Recall $m$ is the number of measurements, and $n$ is the number of pixels in the ground truth image. This measurement process is standard practice in compressed sensing literature; hence we use it on each dataset. Additionally we use a Fourier measurement process common in MRI applications (Mardani et al., 2018; 2017a; Hammernik et al., 2018; Lehtinen et al., 2018; Lustig et al., 2008).

**Datasets:** We use our algorithm to reconstruct both grayscale and RGB images. For grayscale we use the first 100 images in the test set of MNIST (LeCun et al., 1998) and also 60 random images from the Shenzhen Chest X-Ray Dataset (Jaeger et al., 2014), downsampling a 512x512 crop to 256x256 pixels. For RGB we use retinopathy images from the STARE dataset (Hoover et al., 2000) with 512x512 crops downsized to 128x128 pixels.

Table 1: Evaluating the benefits of learned regularization (LR) on x-ray images with varying levels of noise and number of measurements. Table values are percent decrease in error, e.g. at $\sigma_\eta^2 = 0$ and $m = 500$, LR reduces MSE by 9.9%. The term $\sigma_\eta^2$ corresponds to variance of the noise vector $\eta$ in Eqn. 1, i.e. each entry of $\eta$ is drawn independently $\mathcal{N}(0, \frac{\sigma_\eta^2}{m})$. These results indicate that LR tends to provide greater benefit with noisy signals and with fewer measurements.

| | Measurements, $m$ | | | | |
|---|---|---|---|---|---|
| $\sigma_\eta^2$ | 500 | 1000 | 2000 | 4000 | 8000 |
| 0 | 9.9% | 2.9% | 0.2% | 2.0% | 0.6% |
| 10 | 11.6% | 4.6% | 4.5% | 2.4% | 1.0% |
| 100 | 14.9% | 19.2% | 5.0% | 3.9% | 2.8% |
| 1000 | 37.4% | 30.6% | 19.8% | 3.0% | 6.2% |

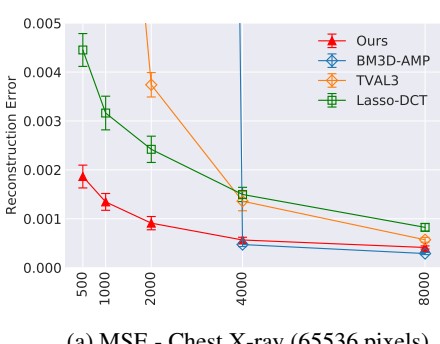

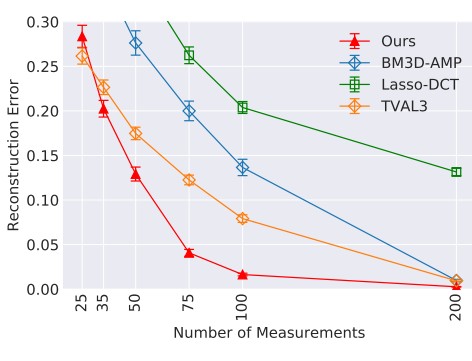

(a) MSE - Chest X-ray (65536 pixels)

(b) MSE - MNIST (784 pixels)

Figure 1: Per-pixel reconstruction error (MSE) vs. number of measurements. Vertical bars indicate 95% confidence intervals. BM3D-AMP frequently fails to converge for fewer than 4000 measurements on x-ray images, as denoted by error values far above the vertical axis.

**Baselines:** We compare our algorithm to state-of-the-art unlearned methods such as BM3D-AMP (Metzler et al., 2016; 2015), TVAL3 (Li, 2011; Li et al., 2009; Zhang et al., 2013), and Lasso in a DCT basis (Ahmed et al., 1974). We also evaluated the performance of Lasso in a Daubechies wavelet basis (Daubechies, 1988; Wasilewski, 2010) but found this performed worse than Lasso - DCT on all datasets. Thus hereon we refer to Lasso - DCT as "Lasso" and do not include results of Lasso - Wavelet.

**Metrics:** To quantitatively evaluate the performance of our algorithm, we use per-pixel mean-squared error (MSE) between the reconstruction $\hat{x}$ and true image $x^*$, i.e. $\frac{\|\hat{x}-x^*\|^2}{n}$. Note that because these pixels are over the range $[-1, 1]$, it's possible for MSE to be greater than 1.

**Implementation:** To find a set of weights $w^*$ that minimize Eqn. 3, we use PyTorch (Paszke et al., 2017) with a DCGAN architecture. For baselines BM3D-AMP and TVAL3, we use the repositories provided by the authors Metzler et al. (2018) and Li et al., respectively. For baseline reconstructions Lasso, we use scikit-learn (Pedregosa et al., 2011). Section A in the appendix provides further details on our experimental procedures, e.g. choosing hyperparameters. The supplementary material also contains our code repository for these experiments.

## 5.2 EXPERIMENTAL RESULTS

### 5.2.1 RESULTS: LEARNED REGULARIZATION

We first evaluate the benefits of learned regularization by comparing our algorithm with and without learned regularization, i.e. $\lambda_L = 100$ and $\lambda_L = 0$, respectively. The latter setting is an unlearned method, as we are not leveraging $(\mu, \Sigma)$ from a specific dataset. In the former setting we first learn $(\mu, \Sigma)$ from a particular set of x-ray images; we then evaluate on a different set of x-ray images. We compare these two settings with varying noise and across different number of measurements.

Our results in Table 1 show that learned regularization does indeed provide benefit. This benefit tends to increase with more noise or fewer measurements. Thus we can infer that assuming a learned Gaussian distribution over weights is useful, especially when the original signal is noisy or significantly compressed.

### 5.2.2 RESULTS: UNLEARNED CS-DIP

For the remainder of this section, we evaluate our algorithm in the noiseless case without learned regularization, i.e. when $\eta = 0$ in Eqn. 1 and $\lambda_L = 0$ in Eqn. 3. Hence CS-DIP is completely unlearned; as such, we compare it to other state-of-the-art unlearned algorithms on various datasets and with different measurement matrices.

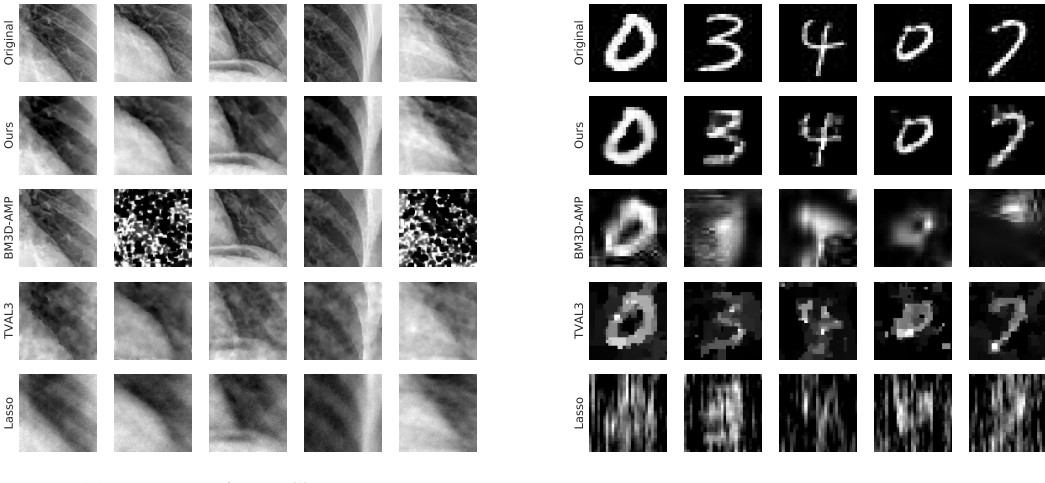

(a) Reconstructions - Chest X-ray

(b) Reconstructions - MNIST

Figure 2: Reconstruction results on x-ray images for m = 2000 measurements (of n = 65536 pixels) and MNIST for m = 75 measurements (of n = 784 pixels). From top to bottom row: original image, reconstructions by our algorithm, then reconstructions by baselines BM3D-AMP, TVAL3, and Lasso. For x-ray images the number of measurements obtained are 3% the number of pixels (i.e. $\frac{m}{n} = .03$), for which BM3D-AMP often fails to converge.

**MNIST:** In Figure 1b we plot reconstruction error with varying number of measurements $m$ of $n$ = 784. This demonstrates that our algorithm outperforms baselines in almost all cases. Figure 2b shows reconstructions for 75 measurements, while remaining reconstructions are in the appendix.

**Chest X-Rays:** In Figure 1a we plot reconstruction error with varying number of measurements $m$ of $n$ = 65536. Figure 2a shows reconstructions for 2000 measurements; remaining reconstructions are in the appendix. On this dataset we outperform all baselines except BM3D-AMP for higher $m$. However for lower $m$, e.g. when the ratio $\frac{m}{n} \leq 3\%$, BM3D-AMP often doesn't converge. This finding seems to support the work of Metzler et al. (2015): BM3D-AMP performs well on higher $m$, e.g. $\frac{m}{n} \geq 10\%$, but recovery at lower sampling rates is not demonstrated.

**Retinopathy:** We plot reconstruction error with varying number of measurements $m$ of $n$ = 49152 in Figure 3a of the appendix. On this RGB dataset we quantitatively outperform all baselines except BM3D-AMP on higher $m$; however, even at these higher $m$, patches of green and purple pixels corrupt the image reconstructions as seen in Figure 10. Similar to x-ray for lower $m$, BM3D-AMP often fails to produce anything sensible. All retinopathy reconstructions are located in the appendix.

**Fourier Measurement Process:** All previous experiments used a measurement matrix $A$ containing Gaussian i.i.d. entries. We now consider the case where the measurement matrix is a subsampled Fourier matrix, as discussed in Section A of the appendix. Our algorithm outperforms baselines on the x-ray dataset, as shown in the appendix Figure 3b.

**Additional Experiments, Appendix A:** In Figure 4a we compare our algorithm against the learned approach of Bora et al. (2017). In Figure 4b we compare our algorithm against baselines in the presence of noise, i.e. when $\eta \neq 0$ in Eqn. 1. We also demonstrate runtimes for all algorithms on the x-ray dataset in Table 2.

# 6 CONCLUSION

We demonstrate how Deep Image Prior (DIP) can be generalized to solve any differentiable linear inverse problem. We further propose a learned regularization method which, with a small amount of prior information, reduces reconstruction error for noisy or compressed measurements. Lastly we prove that the DIP optimization technique can fit any signal given a sufficiently wide single-layer network. This provides theoretical justification for regularization methods such as early stopping.

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

# A  ADDITIONAL EXPERIMENTS, DETAILS, AND INSIGHTS

**Hyperparameters:** Our algorithm CS-DIP is implemented in PyTorch using the RMSProp optimizer (Tieleman & Hinton, 2012) with learning rate $10^{-3}$, momentum 0.9, and 1000 update steps for every set of measurements. These parameters are the same across all datasets. We initialize one random measurement matrix $A$ for each image in all experiments. Further we set the TV hyperparameter $\lambda_T = 0.01$, and found that high frequency components of the image were not reconstructed as sharply with $\lambda_T = 0$.

**Dataset-specific design choices:** On larger images such as x-ray ($n = 65536$) and retinopathy ($n = 49152$), we found no difference using random restarts of the initial seed $z$. However for smaller vectors such as MNIST ($n = 784$), restarts did provide some benefit. As such our experiments utilize 5 random restarts for MNIST and one initial seed (no restarts) for x-ray and retinopathy images. For choosing hyperparameter $\lambda_L$ in Eqn. 3, we used a standard grid search and selected the best one. We used a similar grid search procedure for choosing dataset-specific hyperparameters in baseline algorithms BM3D-AMP, TVAL3, and Lasso.

**Learned Regularization:** For learned regularization, 10 x-ray images are used to learn $\mu$ and $\Sigma$. The results are then evaluated, i.e. averaged over $x^*$, on 50 additional x-ray images.

**Network Architecture:** Our network has depth 7 and uses convolutional layers with ReLU activations. The initial seed $z$ in Eqn. 3 is initialized with random Gaussian i.i.d. entries and then held fixed as we optimize over network weights $w$. We found negligible difference when varying the dimension of $z$ (within reason), as this only affects the number of channels in the network's first layer. Hence we set the dimension of $z$ to be 128, a standard choice for DCGAN architectures.

**Stopping Criterion:** We stop after 1000 iterations in all experiments. We did not optimize this hyperparameter, so admittedly there could be room for improvement. We further note that the "Error vs. Iterations" curve of CS-DIP with RMSProp did not monotonically decrease for some learning rates, even though error gradually decreased in all cases. As such we implemented a stopping condition which chooses the reconstruction with least error over the last 20 iterations. Note we choose this reconstruction based off measurement loss and do not look at the ground truth image.

**Fourier Sampling:** The measurements are obtained by sampling Fourier coefficients along radial lines as demonstrated in Figure 13. For a 2D image $x$ and a set of indices $\Omega$, the measurements we receive are given by $y_{(i,j)} = [\mathcal{F}(x)]_{(i,j)}, (i,j) \in \Omega$, where $\mathcal{F}$ is the 2D Fourier transform. We choose $\Omega$ to be indices along radial lines, as shown in Figure 13 of the appendix; this choice of $\Omega$ is common in literature (Candès et al., 2006) and MRI applications (Mardani et al., 2017a; Lustig et al., 2008; Eksioglu & Tanc, 2018). While Fourier subsampling is common in MRI applications, we use it here on images of x-rays simply to demonstrate that our algorithm performs well with different measurement processes.

We compare our algorithm to baselines on the x-ray dataset for $\{3, 5, 10, 20\}$ radial lines in the Fourier domain, which corresponds to $\{381, 634, 1260, 2500\}$ Fourier coefficients, respectively. We plot reconstruction error with varying number of Fourier coefficients in Figure 3b, outperforming baselines. Reconstructions be found in Figure 14.

Table 2: Runtime (seconds) for each algorithm with varying number of measurements. Note that our algorithm was run on a NVIDIA GTX 1080-Ti GPU, while baselines only leverage CPU. Thus our goal here is not to compare runtimes; instead we demonstrate that our algorithm can run in a reasonable amount of time, which is an issue with other DIP methods.

| Algorithm | 1000 | 2000 | 4000 | 8000 |
|---|---|---|---|---|
| CS-DIP | 15.6 | **17.1** | **20.4** | **29.9** |
| BM3D-AMP | 51.1 | 54.0 | 67.8 | 71.2 |
| TVAL3 | **13.8** | 22.1 | 31.9 | 56.7 |
| Lasso DCT | 27.1 | 33.0 | 52.2 | 96.4 |

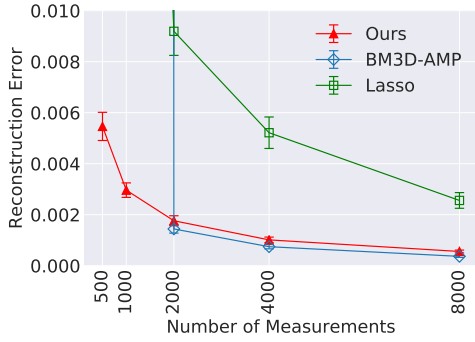 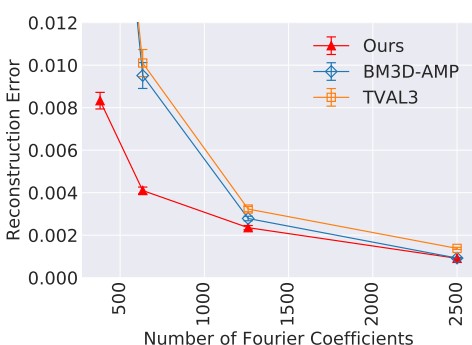

(a) MSE - Retinopathy (RGB) with Gaussian measurements

(b) MSE - Chest X-ray with Fourier measurements

Figure 3: Per-pixel reconstruction error (MSE) vs. number of measurements. Vertical bars indicate 95% confidence intervals. Unfortunately an RGB version of TVAL3 does not currently exist, although related TV algorithms such as FTVd perform similar denoising tasks (Wang et al., 2008).

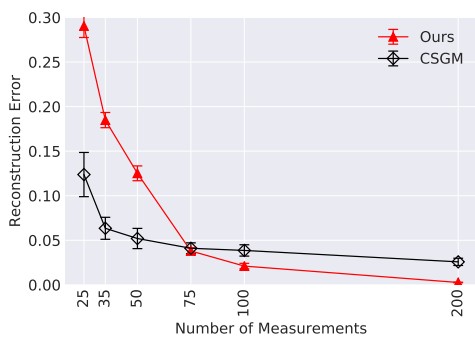 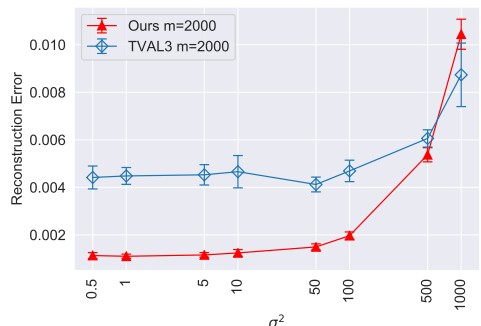

(a) Reconstruction error (MSE) on MNIST for varying number of measurements. As expected, the trained algorithm of Bora et al. (CSGM) outperforms our method for fewer measurements; however, CSGM saturates after 75 measurements, as its output is constrained to the range of the generator. This saturation is discussed in Bora et al., Section 6.1.1.

(b) Reconstruction error (MSE) on x-ray images for varying amounts of noise; number of measurements $m$ fixed at 2000. The term $\sigma^2$ corresponds to variance of the noise vector $\eta$ in $y = Ax + \eta$, i.e. each entry of $\eta$ is drawn independently $\mathcal{N}(0, \frac{\sigma^2}{m})$. Other baselines have error far above the vertical axis and are thus not visible in this plot.

Figure 4

---

**Algorithm 1** Estimate $(\mu, \Sigma)$ for a distribution of optimal network weights $W^*$

**Input:** Set of optimal weights $W^* = \{w_1^*, w_2^*, \cdots, w_Q^*\}$ obtained from $L$-layer DCGAN run over $Q$ images; number of samples $S$; number of iterations $T$.
**Output:** Mean vector $\mu \in \mathbb{R}^L$; covariance matrix $\Sigma \in \mathbb{R}^{L \times L}$.

1: **for** $t = 1$ to $T$ **do**
2:      Sample $q$ uniformly from $\{1, ..., Q\}$
3:      **for** $l = 1$ to $L$ {for each layer} **do**
4:          Get $v \in \mathbb{R}^S$, a vector of $S$ uniformly sampled weights from the $l^{th}$ layer of $w_q^*$
5:          $M_t[l, :] \leftarrow v^T$ where $M_t[l, :]$ is the $l^{th}$ row of matrix $M_t \in \mathbb{R}^{L \times S}$
6:          $\mu_t[l] \leftarrow \frac{1}{S} \sum_{i=1}^{S} v_i$
7:      **end for**
8:      $\Sigma_t \leftarrow \frac{1}{S} M_t M_t^T - \mu_t \mu_t^T$
9: **end for**
10: $\mu \leftarrow \frac{1}{T} \sum_{t=1}^{T} \mu_t$
11: $\Sigma \leftarrow \frac{1}{T} \sum_{t=1}^{T} \Sigma_t$

---

## B PROOF OF SECTION 4: THEORETICAL JUSTIFICATION FOR EARLY STOPPING

In this section we prove our theoretical result in Theorem 4.1. We begin with a summary of some notations we use throughout in Section B.1. Next, we state some preliminary calculations in Section B.2. Then, we state a few key lemmas in Section B.3 with the proofs deferred to Appendix C. Finally, we complete the proof of Theorem 4.1 in Section B.4.

### B.1 NOTATION

In this section we gather some notation used throughout the proofs. We use $\phi(z) = \text{ReLU}(z) = \max(0, z)$ with $\phi'(z) = \mathbb{I}_{\{z \geq 0\}}$. For two matrices/vectors $x$ and $y$ of the same size we use $x \odot y$ to denote the entrywise Hadamard product of these two matrices/vectors. We also use $x \otimes y$ to denote their Kronecker product. For two matrices $B \in \mathbb{R}^{n \times d_1}$ and $C \in \mathbb{R}^{n \times d_2}$, we use the Khatrio-Rao product as the matrix $A = B * C \in \mathbb{R}^{n \times d_1 d_2}$ with rows $A_i$ given by $A_i = B_i \otimes C_i$. For a matrix $M \in \mathbb{R}^{m \times n}$ we use $\text{vect}(M) \in \mathbb{R}^{mn}$ to denote a vector obtained by aggregating the rows of the matrix $M$ into a vector, i.e. $\text{vect}(M) = [M_1 \quad M_2 \quad \ldots \quad M_m]^T$. For a matrix $X$ we use $\sigma_{\min}(X)$ and $\|X\|$ denotes the minimum singular value and spectral norm of $X$. Similarly, for a symetric matrix $M$ we use $\lambda_{\min}(M)$ to denote its smallest eigenvalue.

### B.2 PRELIMINARIES

In this section we carryout some simple calculations yielding simple formulas for the gradient and Jacobian mappings. We begin by noting we can rewrite the gradient descent iterations in the form

$$\text{vect}(W_{\tau+1}) = \text{vect}(W_\tau) - \eta\text{vect}(\nabla\mathcal{L}(W_\tau)).$$

Here,

$$\text{vect}(\nabla\mathcal{L}(W_\tau)) = \mathcal{J}^T(W_\tau)r(W_\tau),$$

where

$$\mathcal{J}(W) = \frac{\partial}{\partial\text{vect}(W)}f(W) \quad \text{and}$$

is the Jacobian mapping associated to the network and

$$r(W) = \phi(V\phi(Wz)) - y.$$

is the misfit or residual vector. Note that

$$\frac{\partial}{\partial\text{vect}(W)}v^T\phi(Wz) = \begin{bmatrix} v_1\phi'(w_1^T z)x^T & v_2\phi'(w_2^T z)x^T & \ldots & v_k\phi'(w_k^T z)x^T \end{bmatrix}$$

$$= (v \odot \phi'(Wx))^T \otimes x^T$$

Thus

$$\mathcal{J}(W) = (V\text{diag}(\phi'(Wz))) * (1z^T),$$

This in turn yields

$$\mathcal{J}(W)\mathcal{J}^T(W) = (V\text{diag}(\phi'(Wz))\text{diag}(\phi'(Wz))V^T) \odot (\|z\|^2 11^T)$$

$$= \|z\|^2 V\text{diag}(\phi'(Wz) \odot \phi'(Wz))V^T \tag{8}$$

### B.3 LEMMAS FOR CONTROLLING THE SPECTRUM OF THE JACOBIAN AND INITIAL MISFIT

In this section we state a few lemmas concerning the spectral properties of the Jacobian mapping, its perturbation and initial misfit of the model with the proofs deferred to Appendix C.

**Lemma B.1** (Minimum singular value of the Jacobian at initialization). *Let $V \in \mathbb{R}^{n \times d}$ and $W \in \mathbb{R}^{d \times k}$ be random matrices with i.i.d. $\mathcal{N}(0, \nu^2)$ and $\mathcal{N}(0, 1)$ entries and define the Jacobian mapping $\mathcal{J}(W) = (V \, diag\,(\phi'(Wz))) * (1z^T)$. Then as long as $d \geq 3828n$,*

$$\sigma_{\min}(\mathcal{J}(W)) \geq \frac{1}{2}\nu\sqrt{d} \|z\|.$$

*holds with probability at least $1 - 2e^{-n}$.*

**Lemma B.2** (Perturbation lemma). *Let $V \in \mathbb{R}^{n \times d}$ be a matrix with i.i.d. $\mathcal{N}(0, \nu^2)$ entries, $W \in \mathbb{R}^{d \times k}$, and define the Jacobian mapping $\mathcal{J}(W) = (V \, diag\,(\phi'(Wz))) * (1z^T)$. Also let $W_0$ be a matrix with i.i.d. $\mathcal{N}(0, 1)$ entries. Then,*

$$\|\mathcal{J}(W) - \mathcal{J}(W_0)\| \leq \nu \|z\| \left(2\sqrt{n} + \sqrt{6\,(2dR)^{\frac{2}{3}} \log\left(\frac{d}{3\,(2dR)^{\frac{2}{3}}}\right)}\right),$$

*holds for all $W \in \mathbb{R}^{d \times k}$ obeying $\|W - W_0\| \leq R$ with probability at least $1 - e^{-n/2} - e^{-\frac{(2dR)^{\frac{2}{3}}}{6}}$.*

**Lemma B.3** (Spectral norm of the Jacobian). *Let $V \in \mathbb{R}^{n \times d}$ be a matrix with i.i.d. $\mathcal{N}(0, \nu^2)$ entries, $W \in \mathbb{R}^{d \times k}$, and define the Jacobian mapping $\mathcal{J}(W) = (V \, diag\,(\phi'(Wz))) * (1z^T)$. Then,*

$$\|\mathcal{J}(W)\| \leq \nu \left(\sqrt{d} + 2\sqrt{n}\right) \|z\|,$$

*holds for all $W \in \mathbb{R}^{d \times k}$ with probability at least $1 - e^{-n/2}$.*

**Lemma B.4** (Initial misfit). *Let $V \in \mathbb{R}^{n \times d}$ be a matrix with i.i.d. $\mathcal{N}(0, \nu^2)$ entries with $\nu = \frac{1}{\sqrt{dn}} \frac{\|y\|}{\|z\|}$. Also let $W \in \mathbb{R}^{d \times k}$ be a matrix with i.i.d. $\mathcal{N}(0, 1)$ entries. Then*

$$\|V\phi(Wz) - y\| \leq 3\|y\|,$$

*holds with probability at least $1 - e^{-n/2} - e^{-d/2}$.*

## B.4 PROOF OF THEOREM 4.1

Consider a nonlinear least-squares optimization problem of the form

$$\min_{\theta \in \mathbb{R}^p} \mathcal{L}(\theta) := \frac{1}{2}\|f(\theta) - y\|^2,$$

with $f : \mathbb{R}^p \mapsto \mathbb{R}^n$ and $y \in \mathbb{R}^n$. Suppose the Jacobian mapping associated with $f$ obeys the following three assumptions.

**Assumption 1.** *Fix a point $\theta_0$. We have that $\sigma_{\min}(\mathcal{J}(\theta_0)) \geq 2\alpha$.*

**Assumption 2.** *Let $\|\cdot\|$ denote a norm that is dominated by the Euclidean norm i.e. $\|\theta\| \leq \|\theta\|$ holds for all $\theta \in \mathbb{R}^p$. Fix a point $\theta_0$ and a number $R > 0$. For any $\theta$ satisfying $\|\theta - \theta_0\| \leq R$, we have that $\|\mathcal{J}(\theta_0) - \mathcal{J}(\theta)\| \leq \alpha/3$.*

**Assumption 3.** *For all $\theta \in \mathbb{R}^p$, we have that $\|\mathcal{J}(\theta)\| \leq \beta$.*

Under these assumptions we can state the following theorem from Oymak & Soltanolkotabi (2019).

**Theorem B.5** (Non-smooth Overparameterized Optimization). *Given $\theta_0 \in \mathbb{R}^p$, suppose Assumptions 1, 2, and 3 hold with*

$$R = \frac{5\|y - f(\theta_0)\|}{\alpha}.$$

*Then, picking constant learning rate $\eta \leq \frac{1}{\beta^2}$, all gradient iterations obey the followings*

$$\|y - f(\theta_\tau)\| \leq (1 - \frac{\eta\alpha^2}{4})^\tau \|y - f(\theta_0)\| \tag{9}$$

$$\frac{\alpha}{5}\|\theta_\tau - \theta_0\| + \|y - f(\theta_\tau)\| \leq \|y - f(\theta_0)\|. \tag{10}$$

We shall apply this theorem to the case where the parameter is $W$ and the nonlinear mapping is given by $V\phi(Wz)$ and $\phi = ReLU$. All that is needed to be able to apply this theorem is check that the assumptions hold. Per the assumptions of the theorem we use

$$\nu = \frac{1}{\sqrt{dn}}\frac{\|y\|}{\|z\|}.$$

To this aim note that using Lemma B.1 Assumption 1 holds with

$$\alpha = \frac{1}{4}\nu\sqrt{d}\|z\| = \frac{1}{4\sqrt{n}}\|y\|,$$

with probability at least $1 - 2e^{-n}$. Furthermore, by Lemma B.3 Assumption 3 holds with

$$\beta = \frac{\|y\|}{\sqrt{8n}}\sqrt{4n+d} \geq \frac{1}{2}\left(\sqrt{\frac{d}{4n}}+1\right)\|y\| = \nu\left(\sqrt{d}+2\sqrt{n}\right)\|z\|.$$

with probability at least $1 - e^{-n/2}$. All that remains for applying the theorem above is to verify Assumption 2 holds with high probability

$$R = 60\sqrt{n} = 15\frac{\|y\|}{\alpha} \geq \frac{5}{\alpha}\|V\phi(Wz) - y\|$$

In the above we have used Lemma B.4 to conclude that $\|V\phi(Wz) - y\| \leq 3\|y\|$ holds with probability at least $1 - e^{-n/2} - e^{-d/2}$. Thus, using Lemma B.2 all that remains is to show that

$$\frac{1}{\sqrt{dn}}\|y\|\left(2\sqrt{n} + \sqrt{6(2dR)^{\frac{2}{3}}\ln\left(\frac{d}{3(2dR)^{\frac{2}{3}}}\right)}\right) \leq \frac{\alpha}{3} = \frac{1}{12\sqrt{n}}\|y\|,$$

holds with $R = 60\sqrt{n}$ and with probability at least $1 - e^{-n/2} - e^{-\frac{(120)^{\frac{2}{3}}}{6}d^{\frac{2}{3}}n^{\frac{1}{3}}} \geq 1 - e^{-n/2} - e^{-4d^{\frac{2}{3}}n^{\frac{1}{3}}}$. The latter is equivalent to

$$2\sqrt{n} + \sqrt{6(120d\sqrt{n})^{\frac{2}{3}}\ln\left(\frac{d}{3(120d\sqrt{n})^{\frac{2}{3}}}\right)} \leq \frac{\sqrt{d}}{12},$$

which can be rewritten in the form

$$2\sqrt{\frac{n}{d}} + \sqrt{6(120)^{\frac{2}{3}}\sqrt[3]{\frac{n}{d}}\ln\left(\frac{1}{3(120)^{\frac{2}{3}}\sqrt[3]{\frac{n}{d}}}\right)} \leq \frac{1}{12},$$

which holds as long as $d \geq 4.3 \times 10^{15}n$. Thus with $d \geq Cn$ then Assumptions 1, 2, and 3 holds with probability at least $1 - 5e^{-n/2} - e^{-d/2} - e^{-4d^{\frac{2}{3}}n^{\frac{1}{3}}}$. Thus, Theorem B.5 holds with high probability. Applying Theorem B.5 completes the proof.

## C  PROOF OF LEMMAS FOR THE SPECTRAL PROPERTIES OF THE JACOBIAN

### C.1  PROOF OF LEMMA B.1

We prove the result for $\nu = 1$, the general result follows from a simple re-scaling. Define the vectors

$$a_\ell = V_\ell\phi'(\langle w_\ell, z\rangle) \in \mathbb{R}^n,$$

with $V_\ell$ the $\ell$th column of $V$. Using equation 8 we have

$$\mathcal{J}(W)\mathcal{J}^T(W) = \|z\|^2 V\text{diag}(\phi'(Wz) \odot \phi'(Wz))V^T,$$

$$= \|z\|^2\left(\sum_{\ell=1}^d a_\ell a_\ell^T\right),$$

$$= d\|z\|^2\left(\frac{1}{d}\sum_{\ell=1}^d a_\ell a_\ell^T\right). \tag{11}$$

To bound the minimum eigenvalue we state a result from Oliveira (2013).

**Theorem C.1.** *Assume $A_1, \ldots, A_d \in \mathbb{R}^{n \times n}$ are i.i.d. random positive semidefinite matrices whose coordinates have bounded second moments. Define $\Sigma := \mathbb{E}[A_1]$ (this is an entry-wise expectation) and*

$$\widehat{\Sigma}_d = \frac{1}{d} \sum_{\ell=1}^{d} A_\ell.$$

*Let $h \in (1, +\infty)$ be such that $\sqrt{\mathbb{E}\left[ (u^T A_1 u)^2 \right]} \leq h u^T \Sigma u$ for all $u \in \mathbb{R}^n$. Then for any $\delta \in (0, 1)$ we have*

$$\mathbb{P}\left\{ \forall u \in \mathbb{R}^n : u^T \widehat{\Sigma}_k u \geq \left( 1 - 7h\sqrt{\frac{n + 2\ln(2/\delta)}{d}} \right) u^T \Sigma u \right\} \geq 1 - \delta$$

We shall apply this theorem with $A_\ell := a_\ell a_\ell^T$. To do this we need to calculate the various parameters in the theorem. We begin with $\Sigma$ and note that for ReLU we have

$$
\begin{aligned}
\Sigma :=& \mathbb{E}[A_1] \\
=& \mathbb{E}\left[ a_1 a_1^T \right] \\
=& \mathbb{E}_{w \sim \mathcal{N}(0, I_k)}\left[ \left( \phi'(\langle w, z \rangle) \right)^2 \right] \mathbb{E}_{v \sim \mathcal{N}(0, I_n)}[vv^T] \\
=& \mathbb{E}_{w \sim \mathcal{N}(0, I_k)}\left[ \left( \phi'(w^T z) \right)^2 \right] I_n \\
=& \mathbb{E}_{w \sim \mathcal{N}(0, I_k)}\left[ \mathbb{I}_{\{w^T z \geq 0\}} \right] I_n \\
=& \frac{1}{2} I_n.
\end{aligned}
$$

To calculate $h$ we have

$$
\begin{aligned}
\sqrt{\mathbb{E}\left[ (u^T A_1 u)^2 \right]} \leq& \sqrt{\mathbb{E}\left[ (a_1^T u)^4 \right]} \\
\leq& \sqrt{\mathbb{E}_{w \sim \mathcal{N}(0, I_k)}\left[ \mathbb{I}_{\{w^T z \geq 0\}} \right] \cdot \mathbb{E}_{v \sim \mathcal{N}(0, I_n)}\left[ (v^T u)^4 \right]} \\
\leq& \sqrt{\frac{3}{2} \|u\|^4} \\
\leq& \frac{\sqrt{3}}{\sqrt{2}} \|u\|^2 \\
=& \sqrt{6} u^T \left( \frac{1}{2} I_n \right) u \\
=& \sqrt{6} \cdot u^T \Sigma u.
\end{aligned}
$$

Thus we can take $h = \sqrt{6}$. Therefore, using Theorem C.1 with $\delta = 2e^{-n}$ we can conclude that

$$\lambda_{\min}\left( \frac{1}{d} \sum_{\ell=1}^{d} a_\ell a_\ell^T \right) \geq \frac{1}{4}$$

holds with probability at least $1 - 2e^{-n}$ as long as

$$d \geq 3528 \cdot n.$$

Plugging this into equation 11 we conclude that with probability at least $1 - 2e^{-n}$

$$\sigma_{\min}\left( \mathcal{J}(W) \right) \geq \frac{1}{2} \sqrt{d} \|z\|.$$

## C.2 PROOF OF LEMMA B.2

We prove the result for $\nu = 1$, the general result follows from a simple rescaling. Based on equation 8 we have

$$\left( \mathcal{J}(W) - \mathcal{J}(W_0) \right) \left( \mathcal{J}(W) - \mathcal{J}(W_0) \right)^T = \|z\|^2 V \mathrm{diag}\left( \left( \phi'(Wz) - \phi'(W_0 z) \right) \odot \left( \phi'(Wz) - \phi'(W_0 z) \right) \right) V^T.$$

Thus

$$\|\mathcal{J}(W) - \mathcal{J}(W_0)\| \leq \|z\| \left\| V \text{diag} \left( \phi'(Wz) - \phi'(W_0 z) \right) \right\| c \tag{12}$$

$$= \|z\| \left\| V \text{diag} \left( \mathbb{I}_{\{Wz \geq 0\}} - \mathbb{I}_{\{W_0 z \geq 0\}} \right) \right\|$$

$$\leq \|z\| \left\| V_{\mathcal{S}(W)} \right\|, \tag{13}$$

where $\mathcal{S}(W) \subset \{1, 2, \ldots, d\}$ is the set of indices where $Wz$ and $W_0 z$ have different signs i.e. $\mathcal{S}(W) := \{\ell : \text{sgn}(e_\ell^T W z) \neq \text{sgn}(e_\ell^T W_0 z)\}$ and $V_{\mathcal{S}(W)}$ is a submatrix $V$ obtained by picking the columns corresponding to $\mathcal{S}(W)$.

To continue further note that by Gordon's lemma we have

$$\sup_{|\mathcal{S}| \leq s} \|V_{\mathcal{S}}\| \leq \sqrt{n} + \sqrt{2s \log(d/s)} + t,$$

with probability at least $1 - e^{-t^2/2}$. In particular using $t = \sqrt{n}$ we conclude that

$$\sup_{|\mathcal{S}| \leq s} \|V_{\mathcal{S}}\| \leq 2\sqrt{n} + \sqrt{2s \log(d/s)}, \tag{14}$$

with probability at least $1 - e^{-n/2}$. To continue further we state a lemma controlling the size of $|\mathcal{S}(W)|$ based on the size of the radius $R$.

**Lemma C.2** (sign changes in local neighborhood). *Let $W_0 \in \mathbb{R}^{d \times k}$ be a matrix with i.i.d. $\mathcal{N}(0, 1)$ entries. Also for a matrix $W \in \mathbb{R}^{d \times k}$ define $\mathcal{S}(W) := \{\ell : sgn(e_\ell^T W z) \neq sgn(e_\ell^T W_0 z)\}$. Then for any $W \in \mathbb{R}^{d \times k}$ obeying $\|W - W_0\| \leq R$*

$$|\mathcal{S}(W)| \leq 2 \lceil (2dR)^{\frac{2}{3}} \rceil$$

*holds with probability at least $1 - e^{-\frac{(2dR)^{\frac{2}{3}}}{6}}$.*

Combining equation 12 together with equation 14 (using $s = 3 (2dR)^{\frac{2}{3}}$) and Lemma C.2 we conclude that

$$\|\mathcal{J}(W) - \mathcal{J}(W_0)\| \leq \|z\| \left( 2\sqrt{n} + \sqrt{6 (2dR)^{\frac{2}{3}} \log \left( \frac{d}{3 (2dR)^{\frac{2}{3}}} \right)} \right)$$

holds with probability at least $1 - e^{-n/2} - e^{-\frac{(2dR)^{\frac{2}{3}}}{6}}$.

## C.3 PROOF OF LEMMA C.2

To prove this result we utilize two lemmas from Oymak & Soltanolkotabi (2019). In these lemmas we use $|v|_{m-}$ to denote the $m$th smallest entry of $v$ after sorting its entries in terms of absolute value.

**Lemma C.3.** *(Oymak & Soltanolkotabi, 2019, Lemma C.2) Given an integer $m$, suppose*

$$\|W - W_0\| \leq \sqrt{m} \frac{|W_0 z|_{m-}}{\|z\|},$$

*then*

$$|\mathcal{S}(W)| \leq 2m.$$

**Lemma C.4.** *(Oymak & Soltanolkotabi, 2019, Lemma C.3) Let $z \in \mathbb{R}^k$. Also let $W_0 \in \mathbb{R}^{d \times k}$ be a matrix with i.i.d. $\mathcal{N}(0, 1)$ entries. Then, with probability at least $1 - e^{-\frac{m}{6}}$,*

$$\frac{|W_0 z|_{m-}}{\|z\|} \geq \frac{m}{2d}.$$

Combining the latter two lemmas with $m = \lceil (2dR)^{\frac{2}{3}} \rceil$ we conclude that when

$$\|W - W_0\| \le R$$
$$\le \frac{m^{\frac{3}{2}}}{2d}$$
$$\le \sqrt{m}\frac{m}{2d}$$
$$\le \sqrt{m}\frac{|W_0 z|_{m-}}{\|z\|},$$

then with probability at least $1 - e^{-\frac{(2dR)^{\frac{2}{3}}}{6}}$ we have

$$|\mathcal{S}(W)| \le 2m \le 2\lceil (2dR)^{\frac{2}{3}} \rceil.$$

## C.4    PROOF OF LEMMA B.3

We prove the result for $\nu = 1$, the general result follows from a simple rescaling. Using equation 8 we have

$$\mathcal{J}(W)\mathcal{J}^T(W) = \|z\|^2 V \text{diag}(\phi'(Wz) \odot \phi'(Wz)) V^T$$

Thus

$$\|\mathcal{J}(W)\| \le \|z\| \|V \text{diag}(\phi'(Wz))\|$$
$$\le \|z\| \|V\|$$

The proof is complete by using standard concentration results for the spectral norm of a Gaussian matrix that allow us to conclude that

$$\|V\| \le \sqrt{d} + 2\sqrt{n},$$

holds with probability at least $1 - e^{-n/2}$.

## C.5    PROOF OF LEMMA B.4

By the triangular inequality we have

$$\|V\phi(Wz) - y\| \le \|V\phi(Wz)\| + \|y\| \tag{15}$$

To continue further let us consider one entry of $V\phi(Wz)$ and note that it has the same distribution as

$$V\phi(Wz) \sim \nu \|\phi(Wz)\| g,$$

where $g \in \mathbb{R}^d$ is random Gaussian vectors with distribution $g \sim \mathcal{N}(0, I_d)$. Thus

$$\|V\phi(Wz)\| \sim \nu \|\phi(Wz)\| \|g\| \le \sqrt{2n}\nu \|\phi(Wz)\| \le \sqrt{2n}\nu \|Wz\|, \tag{16}$$

with probability at least $1 - e^{-n/2}$. Furthermore, note that

$$Wz \sim \|z\| \widetilde{g},$$

where $\widetilde{g} \in \mathbb{R}^d$ is random Gaussian vectors with distribution $\widetilde{g} \sim \mathcal{N}(0, I_d)$. Combining the latter with equation 16 we conclude that

$$\|V\phi(Wz)\| \le 2\sqrt{nd}\nu \|z\| = 2\|y\|,$$

holds with probability at least $1 - e^{-n/2} - e^{-d/2}$. Combining the latter with equation 15 we conclude that

$$\|V\phi(Wz) - y\| \le 3\|y\|,$$

holds with probability at least $1 - e^{-n/2} - e^{-d/2}$.

# D  RECONSTRUCTIONS

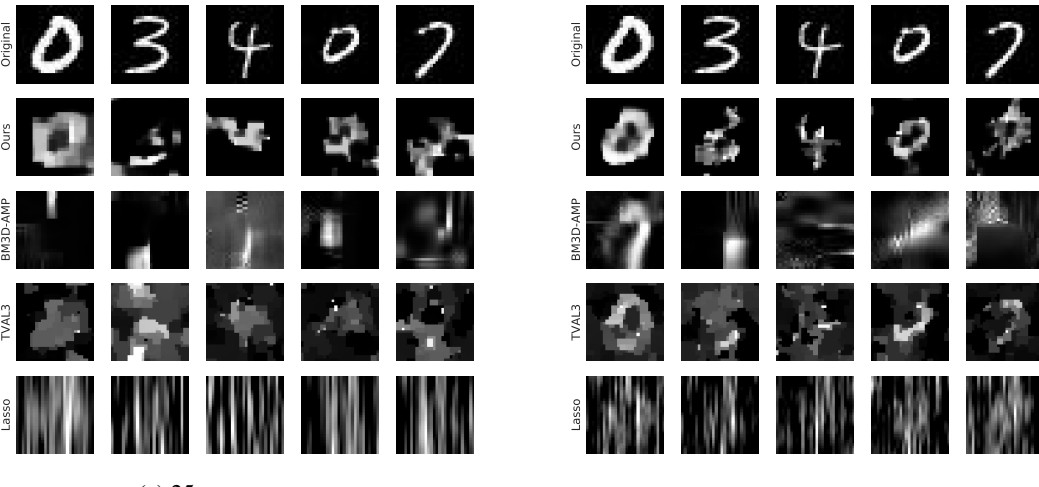

(a) 25 measurements

(b) 50 measurements

Figure 5: Reconstruction results on MNIST for m = 25, 50 measurements respectively (of n = 784 pixels). From top to bottom row: original image, reconstructions by our algorithm, then reconstructions by baselines BM3D-AMP, TVAL3, and Lasso.

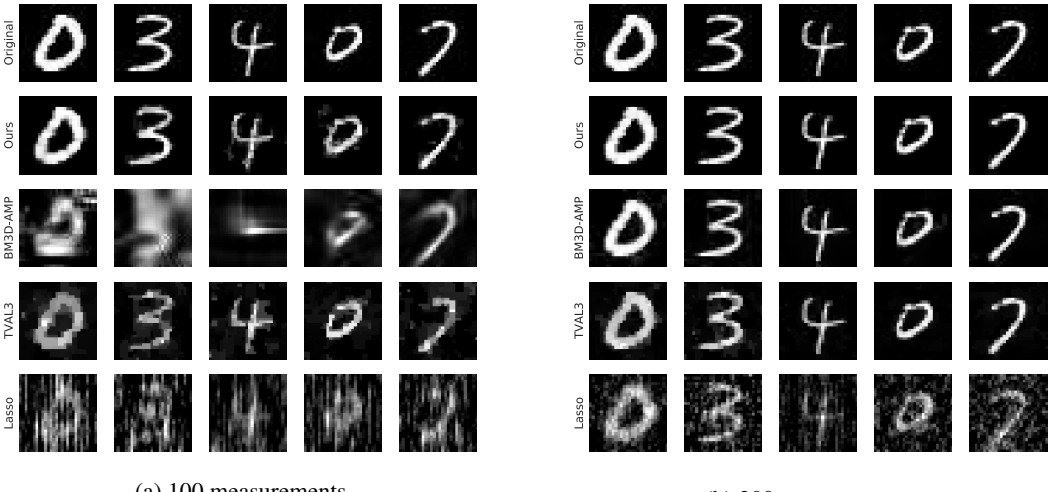

(a) 100 measurements

(b) 200 measurements

Figure 6: Reconstruction results on MNIST for m = 100, 200 measurements respectively (of n = 784 pixels). From top to bottom row: original image, reconstructions by our algorithm, then reconstructions by baselines BM3D-AMP, TVAL3, and Lasso.

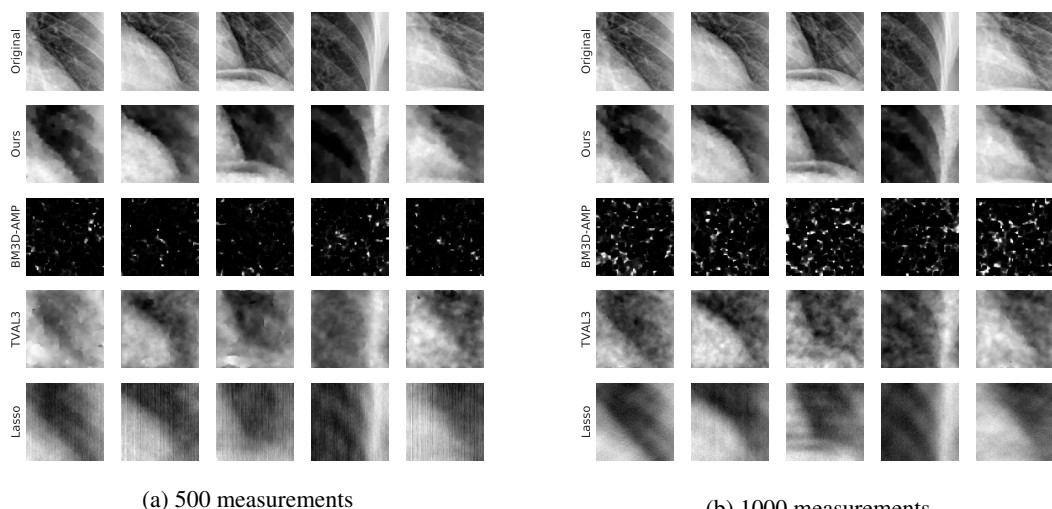

(a) 500 measurements

(b) 1000 measurements

Figure 7: Reconstruction results on x-ray images for m = 500, 1000 measurements respectively (of n = 65536 pixels). From top to bottom row: original image, reconstructions by our algorithm, then reconstructions by baselines BM3D-AMP, TVAL3, and Lasso.

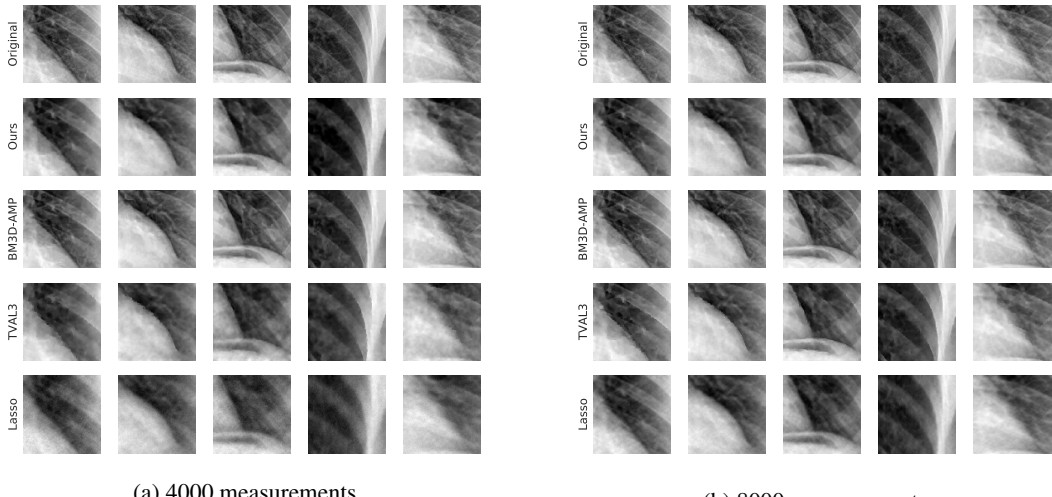

(a) 4000 measurements

(b) 8000 measurements

Figure 8: Reconstruction results on x-ray images for m = 4000, 8000 measurements respectively (of n = 65536 pixels). From top to bottom row: original image, reconstructions by our algorithm, then reconstructions by baselines BM3D-AMP, TVAL3, and Lasso.

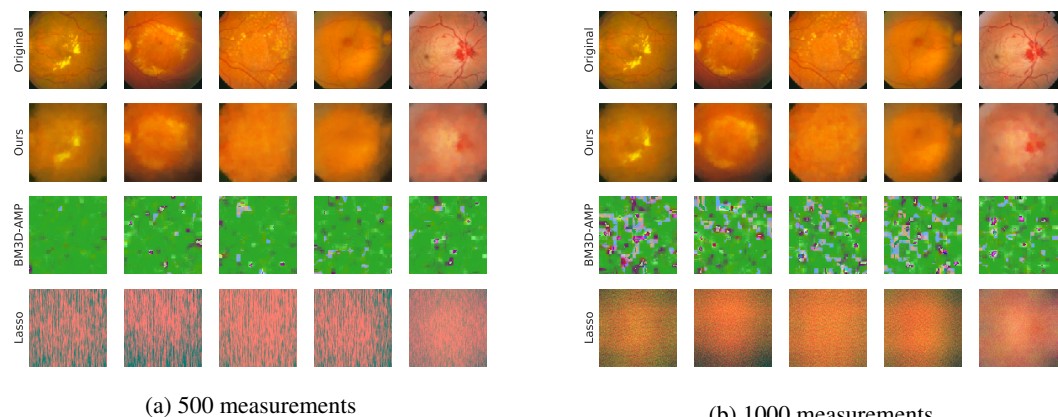

(a) 500 measurements

(b) 1000 measurements

Figure 9: Reconstruction results on retinopathy images for m = 500, 1000 measurements respectively (of n = 49152 pixels). From top to bottom row: original image, reconstructions by our algorithm, then reconstructions by baselines BM3D-AMP and Lasso.

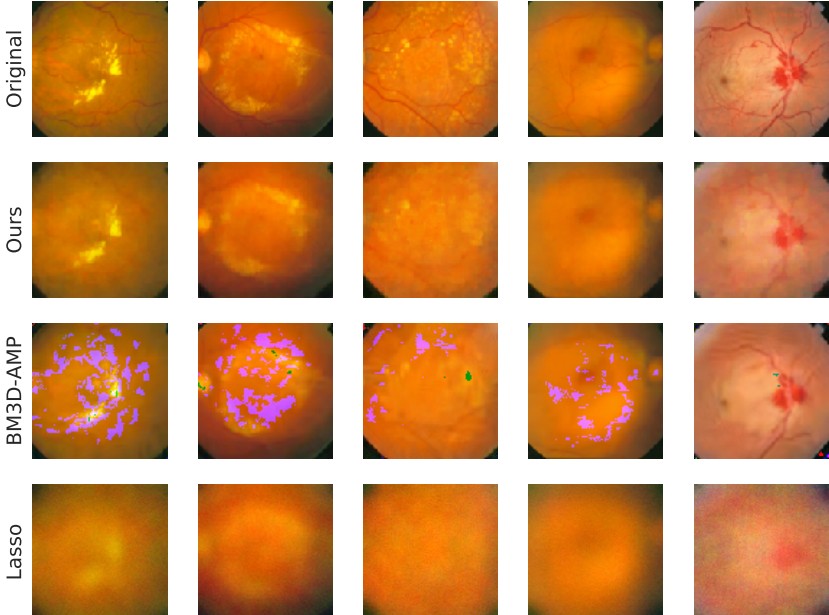

Figure 10: Reconstruction results on retinopathy images for m = 2000 measurements (of n = 49152 pixels). From top to bottom row: original image, reconstructions by our algorithm, then reconstructions by baselines BM3D-AMP and Lasso. In this case the number of measurements is much smaller than the number of pixels (roughly 4% ratio), for which BM3D-AMP fails to converge, as demonstrated by erroneous green and purple pixels. We recommend viewing in color.

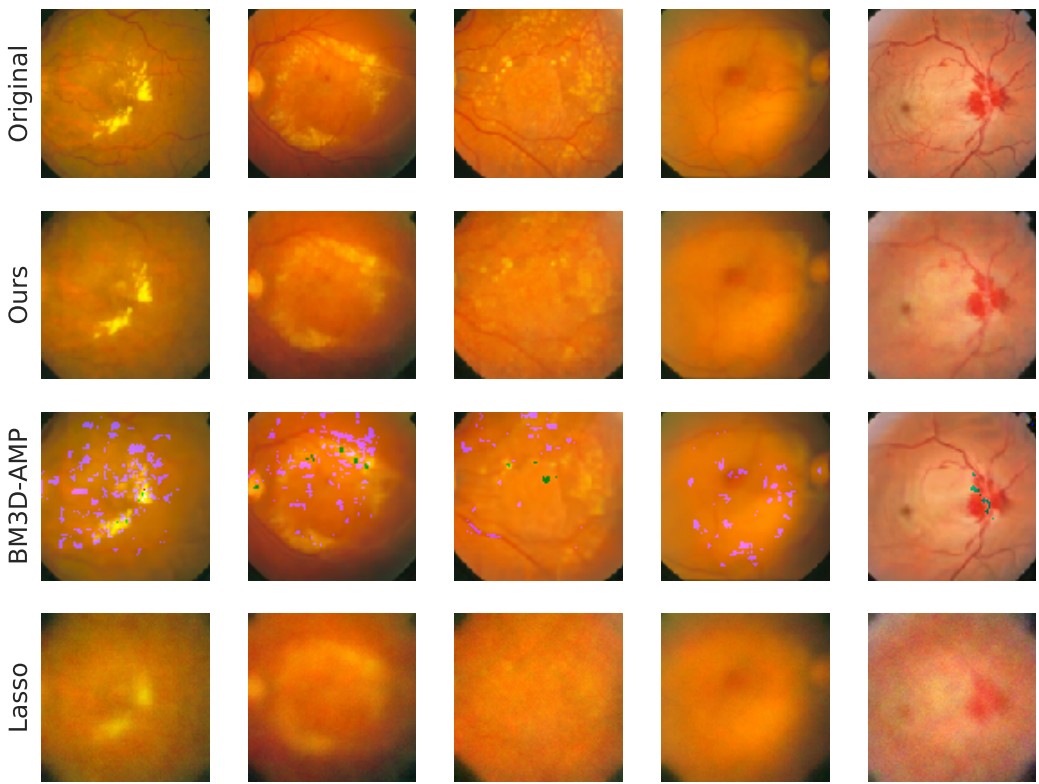

Figure 11: Reconstruction results on retinopathy images for m = 4000 (of n = 49152 pixels). From top to bottom row: original image, reconstructions by our algorithm, then reconstructions by baselines BM3D-AMP and Lasso.

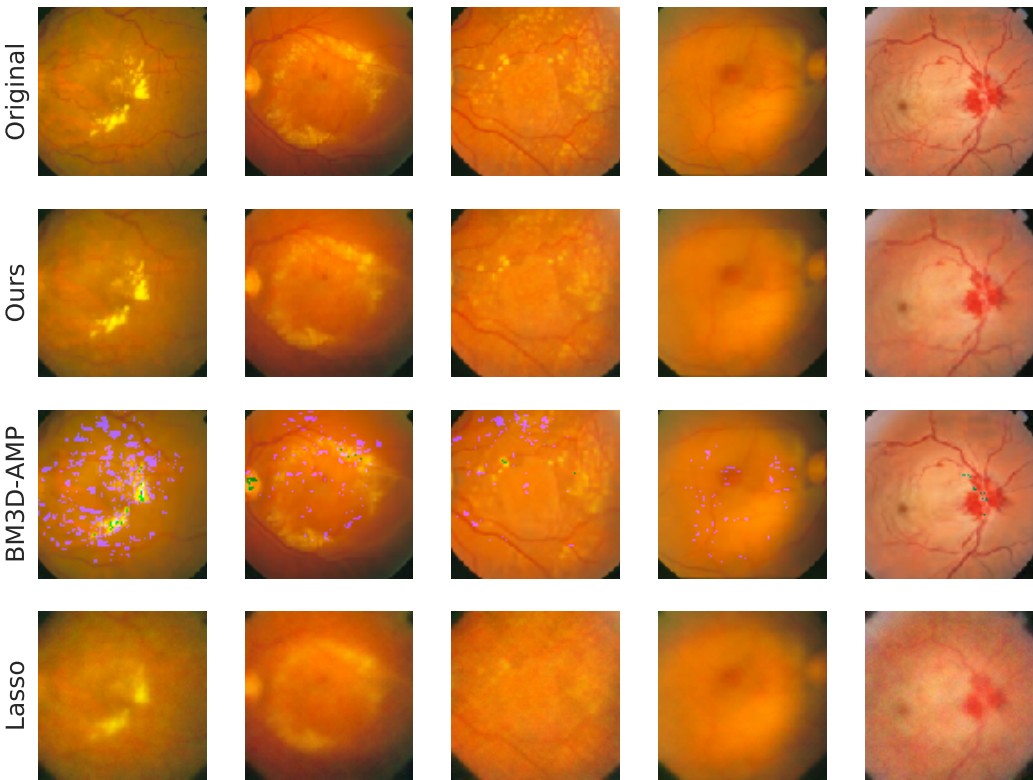

Figure 12: Reconstruction results on retinopathy images for m = 8000 (of n = 49152 pixels). From top to bottom row: original image, reconstructions by our algorithm, then reconstructions by baselines BM3D-AMP and Lasso.

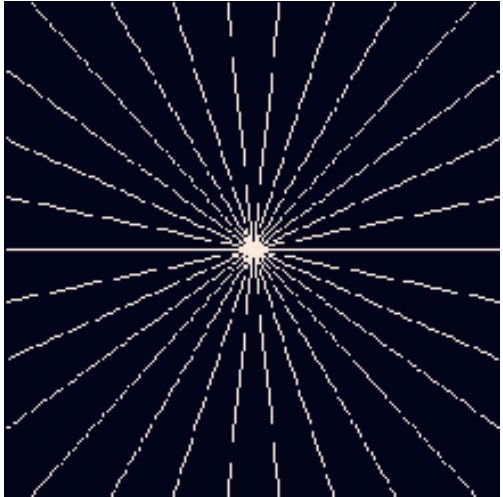

Figure 13: A radial sampling pattern of coefficients $\Omega$ in the Fourier domain. The measurements are obtained by sampling Fourier coefficients along these radial lines.

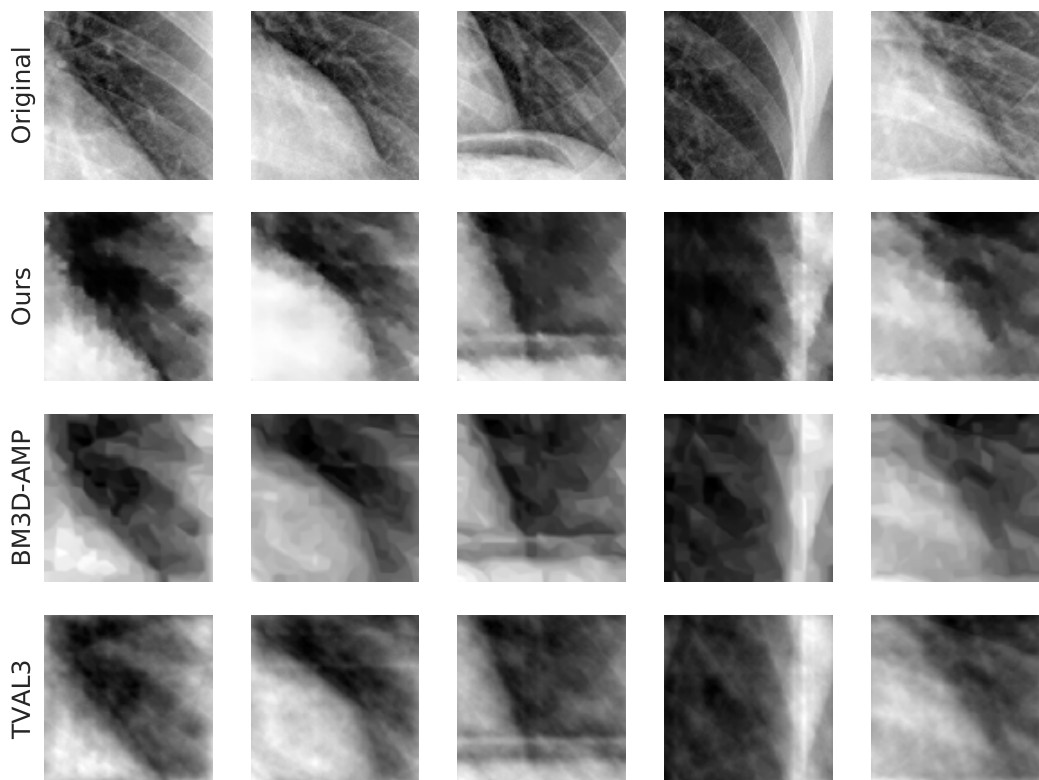

Figure 14: Reconstruction results on x-ray images for m = 1260 Fourier coefficients (of n = 65536 pixels). From top to bottom row: original image, reconstructions by our algorithm, then reconstructions by baselines BM3D-AMP and TVAL3.

