# OpenReview forum: "Compressed Sensing with Deep Image Prior and Learned Regularization"
_ICLR.cc/2020/Conference — Reject_

### Official Review · AnonReviewer2 · 2019-10-24
**Official Blind Review #2**

**Rating:** 6

**Review:**

This paper proposes use of the deep image prior (DIP) in compressed sensing. The proposed method, termed CS-DIP, solves the nonlinear regularized least square regression (equation (3)). It is especially beneficial in that it does not require training using a large-scale dataset if the learned regularization is not used. Results of numerical experiments demonstrate empirical superiority of the proposed method on the reconstruction of chest x-ray images as well as on that of the MNIST handwritten digit images.

The demonstrated empirical efficiency of the proposal is itself interesting. At the same time, the proposal can be regarded as a straightforward combination of compressed sensing and the DIP, so that the main contribution of this paper should be considered rather marginal. I would thus like to recommend "weak accept" of this paper.

In my view the concept of DIP provides a very stimulating working hypothesis which claims that what is important in image reconstruction is not really representation learning but rather an appropriate network architecture. The results of this paper can be regarded as providing an additional empirical support for this working hypothesis. On the other hand, what we have understood in this regard seem quite limited; for example, on the basis of the contents of this paper one cannot determine what network architecture should be used in the proposed CS-DIP framework applied to a specific task. I think that the fact that DIP is still a working hypothesis should be stressed more in this paper. It does not reduce the value of this paper as one providing an empirical support for it.

I think that the theoretical result of this paper, summarized as Theorem 4.1, does not tell us much about CS-DIP. The theorem shows that overfitting occurs even if one uses a single hidden-layer ReLU network. As the authors argue, it would suggest necessity of early stopping in the proposal. On the other hand, I could not find any discussion on early stopping in the experiments: On page 14, lines 22-23, it seems that the theoretical result is not taken into account in deciding the stopping criterion, which would make the significance of the theoretical contribution quite obscure.

Page 4, line 10: a phenomen(a -> on)
Page 13: The paper by Ulyanov, Vedaldi, and Lempitsky on DIP has been published as a conference paper in the proceedings of CVPR 2018, so that appropriate bibliographic information should be provided.
Page 16, line 26: On the right-hand side of the equation, the outermost \phi should not be there.
Page 17, line 3: 3828 should perhaps read 3528.


**Experience Assessment:**

I have published one or two papers in this area.

**Review Assessment: Checking Correctness Of Derivations And Theory:**

I assessed the sensibility of the derivations and theory.

**Review Assessment: Checking Correctness Of Experiments:**

I assessed the sensibility of the experiments.

**Review Assessment: Thoroughness In Paper Reading:**

I read the paper at least twice and used my best judgement in assessing the paper.

---

### Official Review · AnonReviewer1 · 2019-10-26
**Official Blind Review #1**

**Rating:** 6

**Review:**

This paper proposes a new algorithm for compressed sensing using untrained generative model. To the best of my knowledge, the proposed method is new. The closest method, as stated by the authors, is by Bora et al. (2017). However, Bora et al. (2017) is a pre-trained method and would require access to a large number of samples. This paper, in comparison, only requires access to a small number of samples as it is an untrained generative model.

The major theoretical contribution is that by running gradient descent, for a network with single hidden layer, the training error will decrease to 0. This provides theoretical justification for early stopping adopted by the authors.

My complaint to the paper is that the theoretical justification of the generalization ability of the proposed method is missing. The error goes to 0 is not surprising by many existing works, and the authors completely ignore the generalization error analysis of the proposed method.

**Experience Assessment:**

I have published one or two papers in this area.

**Review Assessment: Checking Correctness Of Derivations And Theory:**

I assessed the sensibility of the derivations and theory.

**Review Assessment: Checking Correctness Of Experiments:**

I did not assess the experiments.

**Review Assessment: Thoroughness In Paper Reading:**

I read the paper at least twice and used my best judgement in assessing the paper.

---

### Official Review · AnonReviewer3 · 2019-10-30
**Official Blind Review #3**

**Rating:** 3

**Review:**

== After reading authors' response ==

The authors didn't really offer a response. Their new PDF just mentions that their proof for the case $A=I$ can be extended to any orthogonal matrix $A$ (since if $G$ is a iid Gaussian matrix with unit variance, so is $AG$ for any orthogonal matrix $A$).  Hence, I don't change my response, and still support "weak accept"


== Original review ==
The paper proposes an extension to the 2017 "Deep Image Prior" (DIP) idea. DIP is a method to use untrained neural nets for compressed sensing, but uses the structure of the neural net to implicitly regularize and constrain the solution (though, as a major result of this paper shows, it doesn't regularize -- it can learn noise). It's an interesting combination of compressed sensing and deep learning.

This particular paper adds the following to the DIP idea:

(1) it adds a total-variation (TV) semi-norm regularization term,

(2) it adds a weighted Tikhonov/ridge-regression style regularization term, with learned weight-matrix, and some overall lambda_L hyper-parameter,

(3) they provide a theorem, based on a 2019 preprint, that a very simplified version of the unregularized DIP model (1 layer, with one set of weights fixed and assumed drawn Gaussian) can be fully trained by gradient descent, and hence it can overfit to noise, and thus necessarily must be stopped early.

They provide some numerical experiments to demonstrate the performance of their approach.


== I have some high-level criticism of the approach (things that cannot be fixed):

-- The idea of an algorithm that must be stopped early is distasteful to me and perhaps to others as well. It's vague, and it means the model is wrong. It doesn't feel like math, it feels like sorcery. You haven't actually provided a result that says when to stop, or that stopping early at all will help, only that running until convergence is a bad idea.

-- Compressed sensing is a bit old these days, and part of the greatness of CS was what it inspired, even if single-pixel cameras have not overturned the camera industry. But pure CS, with a Gaussian measurement matrix, and noiseless measurements, is quite academic and not applicable to the real-world.  In particular, your method seems to do best when there are a lot of measurements (for low measurement numbers, the Bora et al approach does well, as does TVAL on the MNIST data -- see Fig 1b). The experiments were also noiseless, and they had Gaussian or the Fourier sampling (which is not random, but rather heavily biased toward DC). I'm not that convinced this is useful for medical imaging.


On the plus side, any approach for medical imaging is welcome, even if the connection is a bit academic. The writing was overall good.  Furthermore, the proof of the theorem seems like it combines quite a bit of good math. It wasn't clear if empirical process theory was used in their proof, or if that's from the 2019 result they cite.


== Some major comments (things that *can* be addressed)

-- Experimentally, I wasn't convinced that the learned regularization (LR) was useful. Table 1 appears to show it is helpful, but this is just saying that some kind of l2 regularization is useful. I would be convinced if you showed that vanilla Tikhonov regularization is not as helpful as your learned l2 regularization.

-- Furthermore, for the benefit of your LR approach, you did a grid search on the lambda_L parameter to choose the best value. Since you never mentioned trainining/validation/testing data, I assume you did this on the final set of data? In that case, you've tuned your hyperparameter to your testing data, and so I don't find the results that convincing. I think this is a big deal.  Or maybe I don't understand what you've done, because the language ".. and selected the best one" was unclear what "best" meant, e.g., if you look at a residual, or if you use ground-truth data.

-- For both your LR and vanilla Tikhonov, you can try selecting lambda_L based on the residual or other non-oracle method. Standard methods include generalized cross-validation, Morozov’s discrepancy principle, and the unbiased predictive risk estimator method.

-- For the results comparing with Bora et al., how much of your performance advantage was due to the TV term? Please re-do Fig 4b with variants of your method with lambda_T=0, lambda_L=0, and both lambdas=0 added to the plot.

-- The handling of the TV term was not discussed. It seems you fed this into PyTorch and hoped for the best. Actually minimizing the usual isotropic 2D TV is tricky, and there are many optimization papers written about it. It's not differentiable, and if you actually converge, then you usually land at points of non-differentiability, so it cannot just be ignored. Sometimes it is smoothed, or a separable but non-isotropic version is used. I think your response to this point is, "but we don't actually run the algorithm to convergence" (since it will overfit), which I still find bizarre (imagine saying that in a paper submitted to an optimization journal!).

== Summary of my opinion ==

I liked the idea, but I think I liked the core idea of DIP more than the extensions that this paper provides. The experiments show enough promising results to make this interesting, but they have not explored the details of the extensions.  The theorem is very complex but quite limited in scope, and maybe would be better as its own (math) result rather than as an add-on to this paper.  The application of CS is a bit limited, and the biggest benefits seem to be in the high-measurement and low or zero-noise case.

Overall, I lean toward weak reject, as I think the regularization parts of the paper would benefit from a major revision of the paper.


== Minor comments ==

- In section 3.1, the description of G is confusing. Writing G(z;w): R^k --> R^n is unclear. You mean G(.;w): R^k --> R^n. And calling z a "seed" confused me, until I realized you meant this is the "input" to the layer.

- statements like "we use early stopping, a phenomena that will be analyzed theoretically in Section 4" are misleading, since you don't analyze early stopping at all. You show that your model doesn't regularize, so that it will fit noise (in a special simplified case), so that running to convergence is a bad idea. This doesn't mean you know when to stop.

- Section 3.2, for p(y|w) and p(w), why not just say these are Gaussian and give the parameters?  In the sentence above eq (4), missing a "log" before "posterior".  The justification for the Gaussian weights (last paragraph on page 4) felt rather weak to me.  Like everyone for the past 60 years, you are adding a l2 regularization term because it's simple and it works fairly well and there are not that many alternatives.

- It took me a while to figure out that Sigma was L x L. The description here was confusing (in section 3.2.1), and it was unclear of Sigma_U was a sub-matrix (and if it was diagonal or a scaled identity) or a scalar.

- Thm 4.1:
-- "V is fixed at random" sounds like it would make a proper probabilist squirm!  The statement of the theorem is that "... holds ... with probability at least ...". You mean that this probability is due to the randomness in V?

-- You have a 1-layer network with 2 sets of weights, but in fact fix one set of weights; and it's not compressed sensing but denoising since A=I. I understand that results in this area are hard, but make sure not to oversell your result earlier in the paper, since this limited setting doesn't actually apply to the main problem you're interested in.

-- "While our proof is for the case of A=I, a similar result can be shown for other measurement matrices."  If that's true, then please show it!

- Table 1: for with and without LR, did you have a TV term for both?

- Section 5.2.1, "thus we can infer that assuming a learned Gaussian distribution over weights is useful." No, I don't think this is an appropriate assumption.  Maybe you can infer that shrinking the estimator toward a fixed vector (e.g., 0), and training to find how much to shrink, makes your solution more robust (a bias-variance tradeoff).  In particular, in Table 1, I'd like to see the improvements for one of these entries as a box plot, to see if the improvement is due to controlling bad outliers, or if actually the bulk is really improved.

- In bibliography, capitalize acronyms like "admm", "amp", "mri", etc.

**Experience Assessment:**

I have published one or two papers in this area.

**Review Assessment: Checking Correctness Of Derivations And Theory:**

I assessed the sensibility of the derivations and theory.

**Review Assessment: Checking Correctness Of Experiments:**

I assessed the sensibility of the experiments.

**Review Assessment: Thoroughness In Paper Reading:**

I read the paper at least twice and used my best judgement in assessing the paper.

---

### Official Review · AnonReviewer5 · 2019-11-01
**Official Blind Review #5**

**Rating:** 6

**Review:**

This paper proposes a deep learning based compressed sensing method, CS-DIP which employs deep image prior algorithm to recovering signals especially for medical images from noisy measurements without pre-learning over large dataset. For further optimization, a learned regularization is leveraged over small training dataset.  Experimental results show that the proposed methods outperformed the others. The theoretical analysis of early stopping is also given. Overall, the writing of this paper is good, the idea is novel and the contribution is sufficient. Here are some of my comments:

1.     The authors claim that CS-DIP is "unlearned", which makes me confusing. Although the latent representation $z$ in Eq.(2) is not learned, the weights $w$ is still learned from training over measurements. I think maybe the description is not accurate of saying that the network $G(z;w)$ is unlearned.

2.  It would be better if the authors can compare CS-DIP with more deep-learning based CS methods. However, the proposed method is only compared with three traditional CS recovery method over several datasets. In Figure 4(a), only one deep-learning based method, CSGM, was compared with over MNIST dataset. More comparison should be conducted over all the datasets to demonstrate its efficiency, especially over medical image datasets.

**Experience Assessment:**

I have published one or two papers in this area.

**Review Assessment: Checking Correctness Of Derivations And Theory:**

I assessed the sensibility of the derivations and theory.

**Review Assessment: Checking Correctness Of Experiments:**

I carefully checked the experiments.

**Review Assessment: Thoroughness In Paper Reading:**

I read the paper at least twice and used my best judgement in assessing the paper.

---

### Decision · Program_Chairs · 2019-12-19

**Decision:**

Reject

**Comment:**

This paper proposes a compressed sensing (CS) method which employs deep image prior (DIP) algorithm to recovering signals for images from noisy measurements using untrained deep generative models.  A novel learned regularization technique is also introduced. Experimental results show that the proposed methods outperformed the existing work. The theoretical analysis of early stopping is also given. All reviewers agree that it is novel to combine the deep learning method with compressed sensing. The paper is well written and overall good. However the reviewers also proposed many concerns about method and the experiments, but the authors gave no rebuttal almost no revisions were made on the paper. I would suggest the author to consider the reviewers' concern seriously and resubmit the paper to another conference or journal.